

# Ankylosing spondylitis and kinesiophobia

Ugur Ertem

Department of Physical Medicine and Rehabilitation, Uludag University, Bursa, Turkey

## ABSTRACT

**Background:** Ankylosing spondylitis (AS) is a chronic rheumatic disease that predominantly affects the axial skeleton, causing pain and functional impairment. Kinesiophobia, or fear of movement, is common in patients with chronic pain conditions and can significantly hinder treatment outcomes. This study aims to assess the level of kinesiophobia in AS patients and explore its relationship with demographic characteristics, disease duration, pain intensity, disease activity, and functional impairment.

**Methods:** This single-center study included 35 AS patients from July 2021 to July 2023. Patient demographics, disease duration, disease activity (BASDAI (Bath Ankylosing Spondylitis Disease Activity Index)), functionality (BASFI (Bath Ankylosing Spondylitis Functional Index)), pain intensity (VAS (Visual Analog Scale)), and kinesiophobia (TSK (Tampa Scale of Kinesiophobia)) were recorded and analyzed. Patients were categorized into low and high kinesiophobia groups based on TSK scores.

**Results:** Of the 35 AS patients, 15 (42.86%) had high kinesiophobia levels (TSK ≥37). Patients with high kinesiophobia had significantly higher BASDAI, BASFI, and VAS scores ($p < 0.001$) compared to those with low kinesiophobia. No significant relationship was found between kinesiophobia and age, gender, or disease duration ($p > 0.05$).

**Conclusion:** High levels of kinesiophobia in AS patients are associated with increased pain, disease activity, and functional impairment. Early interventions targeting kinesiophobia could improve treatment outcomes and patient functionality.

## INTRODUCTION

Ankylosing spondylitis (AS) is a chronic rheumatic disease involving the body's axial skeleton, characterized by chronic inflammatory changes in large joints such as the sacroiliac and hip joints (*Yang et al., 2023*). The prevalence of AS may vary between regions and ethnic groups. A systematic review assessing the prevalence of AS by continent found the mean prevalence per 10,000 people to be 23.8 in Europe, 16.7 in Asia, 31.9 in North America, 10.2 in Latin America, and 7.4 in Africa (*Dean et al., 2014*). The mean age of onset of AS worldwide has been reported to be 26 years. Approximately 80% of patients develop their first symptoms under 30 years of age (*Uslu et al., 2024*). One of the biggest manifestations of AS is that it negatively affects mobility. Especially in the later stages of

Corresponding author
Ugur Ertem,
ugurertem@uludag.edu.tr

the disease, individuals' mobility decreases and their mobilization is restricted. After a while, this situation turns into a more serious situation as individuals gradually begin to restrict their own movements. Because although many new treatment options have been developed for AS, exercise therapy remains the cornerstone of AS treatment (*Kim et al., 2022*; *Tada et al., 2023*; *van der Linden & van der Heijde, 1998*).

Exercise therapy, along with non-steroidal anti-inflammatory drugs (NSAIDs), is prescribed for AS patients either as initial therapy or as adjunctive therapy alongside any other treatment used as initial therapy. Exercise therapy can prevent functional limitations and improve the quality of life in patients with AS (*Pina Goncalves et al., 2023*; *Zaggelidou et al., 2023*). In this respect, movement and exercise are indispensable for AS patients.

Kinesiophobia, also known as fear of movement, is defined as an excessive, irrational, and debilitating fear of performing a physical movement due to a feeling of vulnerability to a painful injury or re-injury (*Kori, 1990*; *Luque-Suarez, Martinez-Calderon & Falla, 2019*). An exaggerated negative cognitive and emotional response to anticipated or actual pain is referred to as pain catastrophizing. Additionally, AS patients with long-term pain enter a vicious circle in which chronic pain and functional disabilities trigger each other, their condition paradoxically worsens, and their functional and social lives are more adversely affected (*Asiri et al., 2021*; *Ishak, Zahari & Justine, 2017*; *Miller et al., 2020*). In this context, kinesiophobia further aggravates existing diseases, especially those that progress with chronic pain. Considering that AS is a disorder accompanied by chronic pain and that exercise and movement are among the main components of AS treatment, it would not be difficult to predict that kinesiophobia will further exacerbate AS. *Oskay et al. (2017)* reported that kinesiophobia increases the severity of pain in AS patients and negatively affects the patients' functional status and quality of life. Similarly, *Luque-Suarez, Martinez-Calderon & Falla (2019)* reported that high kinesiophobia levels increased pain intensity in patients with chronic musculoskeletal pain, were associated with a higher rate of disability, and negatively affected the quality of life.

Determining the level of kinesiophobia in patients diagnosed with AS and taking the necessary precautions to prevent kinesiophobia is one of the most important cornerstones for the success of the treatment of the disease. In this disease, which has limited mobility as one of its most important manifestations, exercise therapy and prevention of kinesiophobia are essential for a successful treatment. Therefore, we conducted this study to determine the level of kinesiophobia in patients diagnosed with AS and to determine whether there is a relationship between kinesiophobia and demographic characteristics, disease duration, pain severity, disease activity and level of functional impairment. We believe that the study results will contribute to strategies to prevent kinesiophobia in AS patients.

## MATERIALS AND METHODS

This study was designed as a single-center study. The study protocol was approved by the Bursa Uludağ University Faculty of Medicine Clinical Research Ethics Committee prior to the conduct of the study (Decision No: 2021-6/60, Date: May 26[th], 2021). The study was

conducted in accordance with the ethical considerations outlined in the Declaration of Helsinki. Written informed consent was obtained from all patients included in the study.

The study population consisted of AS patients who were consulted to us from different clinics for various reasons or who were regularly followed up in our outpatient clinic between July 2021 and July 2023. Minimum sample size calculation was made and it was determined that a minimum of 15 patients were required for each group (low kinesiophobia, high kinesiophobia). All patients who applied to the outpatient clinic within the specified time period and met the inclusion criteria were included in the study. During this process, a total of 41 patients applied and 35 met the criteria and agreed to participate in the study. This number was considered sufficient for study planning. Diagnosis of AS was made based on modified New York criteria (*Rudwaleit et al., 2009*). While patients who were older than 18 years of age and who had AS for longer than 1 year were included in the study, patients who had a history of surgery within the last 3 months and who had comorbidities with uncontrolled chronic pain, especially fibromyalgia, chronic fatigue syndrome, anxiety disorders and other psychiatric disorders, were excluded from the study.

The patients' age, gender, and disease duration were recorded on a patient evaluation form. Patients' kinesiophobia level was evaluated with the Tampa Scale of Kinesiophobia (TSK), disease activity with the Bath Ankylosing Spondylitis Disease Activity Index (BASDAI), functionality with the Bath Ankylosing Spondylitis Functional Index (BASFI), and pain intensity with the Visual Analog Scale (VAS).

TSK is a commonly used tool to evaluate the kinesiophobia level. The reliability studies of the Turkish version of TSK were completed. TSK consists of 17 items. Each item is assigned a score between 1 and 4 points. Hence, the total score from TSK ranges between 17 and 68 points. The higher the total score, the higher the kinesiophobia level. A total score of 37 and above indicates a high level of kinesiophobia, and a score below 37 indicates a low level of kinesiophobia. In line with the literature, the threshold TSK score between low and high kinesiophobia levels was deemed 37 points (*Kocyigit & Akaltun, 2020*; *Tunca et al., 2011*; *Vlaeyen et al., 1995*). Accordingly, the patients were divided into two groups: patients with a total TSK score <37, *i.e.*, low kinesiophobia group, and patients with a total TSK score ≥37, *i.e.*, high kinesiophobia group.

BASDAI is a widely used 6-question tool to evaluate disease activity. The reliability studies of the Turkish version of BASDAI were completed. BASDAI evaluates the level of weakness and fatigue, the severity of pain and swelling in the spine and other joints, the degree of sensitivity to touch, especially in the enthesis areas, the degree of discomfort after waking up, and the duration of morning stiffness. Within the scope of BASDAI, patients are asked to rate the severity of the said symptoms they experienced in the last week by marking them on a 10 cm long scale. The average of the scores obtained from the last two questions, which assess the discomfort and stiffness after waking up, is taken and summed with the scores of the other four questions. The BASDAI score is obtained by dividing the total score by 5. BASDAI scores of 4 and above indicate that the patient is in the active phase of AS (*Ay et al., 2004*; *Calin et al., 1999*; *Garrett et al., 1994*; *Karkucak et al., 2010*).

BASFI is a 10-question tool used to evaluate the functional status of patients with AS. The validity and reliability studies of BASFI were completed. BASFI evaluates the degree of difficulty that patients experience in bending over, turning, lying down, standing up from different surfaces, climbing stairs, and activities of daily living. Within the scope of BASFI, patients are asked to rate the degree of difficulty they experienced while performing the said activities by marking them on a 10 cm long scale. The total score is obtained by adding the points given to each question. The BASFI score is obtained by dividing the total score by 10. The higher the BASFI score, the more severe the functional limitations (*Calin et al., 1994*; *Gur Kabul et al., 2021*; *Ozer et al., 2005*).

VAS is a scale used to evaluate the pain intensity of patients. Before using VAS, the meaning of the facial expressions on VAS and what they mean are explained to the patients. While the state of having no pain is assigned 0 points, the most severe pain felt in life, expressed as unbearable pain, is assigned 10 points. The patients' pain intensity in the last week was determined according to VAS (*Arslan et al., 2016*; *Carlsson, 1983*; *Inci & Inci, 2023*).

## Statistical analysis

SPSS 21.0 (Statistical Product and Service Solutions for Windows, Version 21.0; IBM Corp., Armonk, NY, U.S., 2012) software package was used to conduct the statistical analyses of the collected data. The results of the statistical analyses were expressed using descriptive statistics, *i.e.*, mean ± standard deviation values in the case of continuous variables determined to conform to the normal distribution, as median with minimum and maximum values in the case of continuous variables determined not to conform to the normal distribution, and as numbers and percentage values in the case of categorical variables. The normal distribution characteristics of numerical variables were analyzed using the Shapiro-Wilk test. In comparing the differences in numerical variables between two independent groups, the independent samples t-test was used for numerical variables determined to conform to the normal distribution, and the Mann-Whitney U test was used for numerical variables determined not to conform to the normal distribution. Fisher's exact test was used to compare the differences in categorical variables between the groups. The correlations between continuous variables were analyzed using correlation analysis. According to normality test results, Spearman correlation coefficients were calculated. Probability ($p$) statistics of <0.05 were deemed to indicate statistical significance.

## RESULTS

A total of 35 AS patients were included in the study. These patients' demographic characteristics, *i.e.*, age, gender, and disease durations are given in Table 1. In addition the mean TSK scores of the patients and whether they are in the low or high kinesiophobia group, as well as their mean BASDAI, BASFI, and VAS scores, are shown in Table 1. The majority of patients included in the study were male. Low-level kinesiophobia was detected in 57.14% of the patients.

The distribution of patients' demographic characteristics, disease activity, and pain and functionality levels by the kinesiophobia groups is given in Table 2. It was found that the

**Table 1 Distribution of individuals according to clinical evaluation scales.**

|  | $n = 35$* |
|---|---|
| **Tampa Kinesiophobia Score** | 35 (17–63) |
| **Kinesiophobia** | |
| <37 (*Low Kinesiophobia*) | 20 (57.14%) |
| ≥37 (*High Kinesiophobia*) | 15 (42.86%) |
| **BASDAI Score** | 2.90 (1.10–7.20) |
| **BASFI Score** | 3.32 ± 2.34 |
| **VAS Score** | 4 (0–10) |
| **Sex** | |
| Female | 8 (22.86%) |
| Male | **27 (77.14%)** |
| **Age** | 40.77 ± 10.80 |
| **Disease duration** | 10 (2–30) |

Note:
* Data are expressed as $n$ (%), mean ± standard deviation and median (minimum-maximum).

**Table 2 Comparisons between kinesiophobia groups.**

|  | Low Kinesiophobia ($n = 20$)* | High Kinesiophobia ($n = 15$)* | $p$-value |
|---|---|---|---|
| **Age** | 40.45 ± 12.76 | 41.20 ± 7.86 | 0.842[a] |
| **Sex** | | | |
| Female | 6 (30%) | 2 (13.33%) | 0.419[b] |
| Male | 14 (70%) | 13 (86.67%) | |
| **Disease duration** | 9 (2–19) | 12 (5–30) | 0.268[c] |
| **BASDAI Score** | 2.18 ± 0.79 | 6.05 ± 0.80 | **<0.001[a]** |
| **BASFI Score** | 1.73 ± 1.28 | 5.45 ± 1.63 | **<0.001[a]** |
| **VAS Score** | 3 (0–4) | 8 (4–10) | **<0.001[c]** |

Notes:
* Data are expressed as $n$ (%), mean ± standard deviation and median (minimum-maximum).
[a] Independent Sample t Test.
[b] Fisher's Exact Test.
[c] Mann-Whitney U Test.

**Table 3 Correlation between Tampa kinesiophobia scale score and clinical parameters, age and disease duration.**

|  | Tampa kinesiophobia SCALE SCOre | |
|---|---|---|
|  | $r_s$* | $p$-value |
| Age | 0.25 | 0.156 |
| Disease duration | 0.23 | 0.191 |
| BASDAİ Score | 0.91 | **<0.001** |
| BASFİ Score | 0.84 | **<0.001** |
| VAS Score | 0.92 | **<0.001** |

Note:
* $r_s$: Spearman correlation coefficient.

increase in BASDAI, BASFI and VAS scores was associated with a high level of kinesiophobia ($p < 0.001$).

The correlations between patients' TSK scores and age, disease duration, BASDAI, BASFI, and VAS scores are shown in Table 3. In parallel, correlation analysis revealed strong positive correlations between TSK score and BASDAI, BASFI, and VAS scores (r: 0.91, r: 0.84, and r: 0.92, respectively) (*Pripp, 2018*).

## DISCUSSION

In the present study, it was found that the BASDAI, BASFI and VAS scores of AS patients with high kinesiophobia were statistically significantly higher than those of AS patients with low kinesiophobia ($p < 0.001$). This indicates the importance of exercise therapy in AS patients. The present study will include a new perspective to the literature on these findings. It is also important in terms of showing that kinesiophobia is a very important modifying factor in the treatment of AS patients.

Although many new treatment options have been developed for AS, exercise therapy remains the cornerstone of AS treatment. The positive effects of physical exercise and movement on AS are well-established in the literature (*Millner et al., 2016*). It is reported in the literature that exercise therapy increases the quality of life of AS patients, improves their physical condition, and reduces AS-related symptoms (*Liska, 2022*).

In a study investigating the relationship between kinesiophobia and quality of life in patients with AS, *Balevi et al. (2020)* found the mean TSK score of the AS patients to be 40.92 ± 6.65, indicating a high level of kinesiophobia. In another study conducted with 30 AS patients in the Turkish Republic of Northern Cyprus, the mean TSK score of the patients was found to be 38.9 ± 6.7 (*Oksuz et al., 2017*). In comparison, of the 35 patients included in our study, 20 (57.14%) had TSK scores ≥37, *i.e.*, high levels of kinesiophobia, and 15 (42.86%) had TSK scores <37, *i.e.*, low levels of kinesiophobia. From this perspective, we see that our results support the literature.

We found that BASDAI, BASFI, and VAS scores of AS patients with high kinesiophobia were statistically significantly higher than those of AS patients with low kinesiophobia ($p < 0.001$). Along these lines, *Sari et al. (2023)* found that TSK score was correlated with age, disease duration, and BASFI score. In a study evaluating the relationships between kinesiophobia and pain, quality of life, functional status, disease activity, mobility, and depression in AS patients, positive significant relationships were found between TSK score and VAS, BASFI, quality of life, and depression scores. On the other hand, there was no statistically significant correlation between TSK and BASDAI scores, even though BASDAI scores were higher in those with higher TSK scores (*Oskay et al., 2017*). *Selcuk et al. (2018)* found positive relationships between TSK score and VAS-pain (r: 0.684, *p*: 0.000), VAS-fatigue (r: 0,494, *p*: 0.000), BASDAI (r: 0,484, *p*: 0.001) and BASFI (r: 0.389, *p*: 0.008) scores in patients with AS. In general, our study's findings correlate with the studies in the literature evaluating the relationship between AS and kinesiophobia. Today, it is accepted worldwide that exercise therapy is one of the main components of AS treatment. There is ample evidence that exercise improves patients' functional status, disease activity, and

mental state. One of the primary obstacles for patients to exercise is kinesiophobia. Therefore, patients with high kinesiophobia levels are expected to have worse disease activity levels, functional statuses, and pain scores. In our study, as expected, AS patients with high kinesiophobia levels had increased functional dependency, experienced more pain, and had higher disease activity levels. The fact that we did not find a relationship between kinesiophobia level, age, and disease duration may be attributed to our small sample size and the fact that most of our patients were followed for a relatively short period. As a reason, advanced age and long-term pain may create fear of movement in patients. As a matter of fact, several studies, such as *Sari et al.*'s *(2023)* study, have shown that age and disease duration are associated with kinesiophobia (*Altunel Kılınc, Kirmizier & Orucoglu, 2023*). In addition, although the current study has limitations in some respects compared to the existing literature, it makes a very important contribution to the literature in terms of clearly showing the positive correlation between disease activity indices and kinesiophobia level without including individual factors.

The primary limitations of our study were its single-center design and small sample size, which hindered the generalizability of our results. Other limitations of the study include not assessing the medications used by the patients, not using other scale used in AS, and not including smoking and other individual characteristics that may be associated with movement limitations in patients.

## CONCLUSIONS

In conclusion, our findings showed that AS patients had high levels of kinesiophobia and that kinesiophobia negatively affected their treatment. Therefore, treatment success in AS patients depends on taking the necessary precautions before the development of kinesiophobia and early interventions after the development of kinesiophobia. In this context, we think that our study will provide guidance in prescribing exercise therapy to AS patients and contribute to the literature. In addition, our results revealed a strong relationship between BASDAI, BASFI scores, which objectively indicate AS activity, and high kinesiophobia. Kinesiophobia, which is often overlooked by clinicians, is related to treatment success in such an important and disabling disease. The results of the current study will make a significant contribution to the literature in this regard and clearly show the importance of evaluating the level of kinesiophobia at every stage of treatment, including treatment planning and changes in treatment regimens in such patients. If we look at the issue from the patients' perspective, the study results indicate that reducing the level of kinesiophobia will also be effective in pain palliation. In this respect, it is revealed that kinesiophobia should be carefully considered when evaluating the relationship between AS and pain in the literature.

### Funding
The authors received no funding for this work.

## Competing Interests

The authors declare that they have no competing interests.

## Author Contributions

- Ugur Ertem conceived and designed the experiments, performed the experiments, analyzed the data, prepared figures and/or tables, authored or reviewed drafts of the article, and approved the final draft.

## Human Ethics

The following information was supplied relating to ethical approvals (*i.e.*, approving body and any reference numbers):

Bursa Uludağ University Faculty of Medicine Clinical Research Ethics Committee

## Data Availability

The raw data is available in the Supplemental File.

## Supplemental Information

Supplemental information for this article can be found online at http://dx.doi.org/10.7717/peerj.19034#supplemental-information.

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
