# Peer review of "Ankylosing spondylitis and kinesiophobia"

_PeerJ, doi:10.7717/peerj.19034_

## Round 0.1 · original submission · Major Revisions

Please address the comments of the reviewers in a comprehensive revision and in particular, be sure to address the concerns of Reviewer 1 regarding the inclusion of calculations of sample size and effect size.

Reviewer 1 ·

Basic reporting

This article's language is clear and well-written.
The article include sufficient introduction.
However, the background is insufficient. Examining the literature reveals numerous studies examining the presence of kinesiophobia and related health parameters in patients with axial spondyloarthritis.

Experimental design

There is no research question or knowledge gap.
Calculations of sample size or effect size were not made in the study. This is a major metadological shortcoming. It casts a shadow on the validity of the data obtained from the study.

Validity of the findings

Calculations of sample size or effect size were not made in the study. This is a major metadological shortcoming. It casts a shadow on the validity of the data obtained from the study.

Cite this review as

Reviewer 2 ·

Basic reporting

The manuscript was clear and "too the point". There was no points of what seemed unnecessary information. The background information was clear and made the purpose of the study known.

Experimental design

Overall, the research question and hypothesis are clear. The methods for how subject were obtained were mostly clear, I would like more information specifically on the exclusion criteria. "...comorbidities with uncontrolled chronic pain, especially fibromyalga...", what other comorbidities?

Validity of the findings

The finding seem valid and the data to back up the results is provided. The authors include the strengths of the study, as well as their limitations.

Cite this review as

Reviewer 3 ·

Basic reporting

No comment.

Experimental design

No comment.

Validity of the findings

No comment.

Additional comments

The aim of this pilot study for assessing the level of kinesiophobia in AS patients. The paper is clearly written and overall, well presented. There are some points to be considered:
-Introduction section, line 42-43: This information you mentioned; ''The proportion of AS patients presenting when they are over 45 years of age is less than 5%.'' in the introduction is old. A recent study showed that the proportion of AS patients presenting over 45 years of age was 9.9%. (PMID: 38902436) I suggest you correct the information along with the reference.
-Introduction section, line 62: as→AS?
-Methods section, line 86: ''Diagnosis of AS was made based on modified New York criteria.'' Add a reference.
-Methods section: Were patients with psychiatric disease excluded from the study? Please specify.
-Statistical analysis: I suggest that you specify the reference on the basis of which you have chosen the correlation strength.
-Results section, line 146-153: The results section is short. I suggest you explain the important findings in words. Also, please indicate the strength of the correlations (weak, moderate or strong correlation). I suggest you provide a reference to the correlation strength in the statistics section.
-Tables: Combine tables 1 and 2 into one table. Since the number of patients is small in demographic and clinical data, it would be more accurate to give the median (min./max.) value instead of the mean value.
-Discussion section: Start with the main findings of your study. In the first paragraph of the discussion, the results of the study, the intended message and its contribution to the literature (what is new?) should be stated. You should then discuss the researchers' work in relation to what is already known and whether it is compatible with your results. If your new findings are really new, you need to explain them well, usually using more than one reference (remember that references should be as recent as possible).
-Discussion section, line 169-172: ‘’We found that BASDAI, BASFI, and VAS scores of AS patients with high kinesiophobia were statistically significantly higher than those of AS patients with low kinesiophobia (p<0.001). In parallel, correlation analysis revealed strong positive correlations between TSK score and BASDAI, BASFI, and VAS scores (r: 0.91, r: 0.84, and r: 0.92, respectively).’’ Please remove this statement from the discussion section and add it to the results section.

Cite this review as

Reviewer 4 ·

Basic reporting

The study was evaluated as sufficient in terms of use of English, grammar and punctuation. Tables are not well designed. The content of tables 1 and 2 is very limited.

Experimental design

The points reported as limitations in the study (insufficient demographic data of the patients, insufficient number of activity scores, small number of patients, etc.) make the study insufficient for the journal.

Validity of the findings

The weakest aspect of the study is its low original value. It was observed that previous studies on similar subjects had appeared in the literature with larger numbers of patients and data.

Annotated reviews are not available for download in order to protect the identity of reviewers who chose to remain anonymous.
Cite this review as

---

## Round 0.2 · Major Revisions

In the first review round, Reviewer 4 had provided a detailed PDF document with their comments, however you only responded to the few sentences of their 'free text' comments which were provided in the system. Therefore, it seems you may have missed their PDF of comments. As a result, they have re-uploaded their PDF from the 1st round, and provided additional comments below. Please respond to all their comments in a revision.

Reviewer 2 ·

Basic reporting

Clear and concise language, with good grammar. Structured well with shared data. Provided all relevant background information. Provided relevant results to the hypotheses posed.

Experimental design

The research question is well defined with relevant research to support. The results answer the original question. The methods are very well defined and reproducible.

Validity of the findings

The findings seem to be valid. The authors come to a logical conclusion with the data to support, and they understand the limitations of the research. All of the underlying data is provided in the 3 presented tables.

Cite this review as

Reviewer 3 ·

Basic reporting

No comment.

Experimental design

No comment.

Validity of the findings

No comment.

Additional comments

Thank you for modifying your manuscript according to my criticisms.

Cite this review as

Reviewer 4 ·

Basic reporting

Abstract:
In the methods section of the abstract the explanations of the abbreviations as BASDAI, VAS, BASFI, TSK should be included because they used for the first time.
Introduction:
The introduction section is too long. In the first paragraph of the introduction please focus on the AS, mobility, dysfunction and kinesiofobia instead of the long general information. The sentence structures in the introduction part should be evaluated, and numerous repeated conjunctions should be avoided. The study's goal should be thoroughly and clearly explained in the final section of the Introduction. Concentrate on important research and data that bolster the objectives of the study directly.
Methods:
The potential impact of the work is reduced by concerns for some methodological rigor. There are very few demographic data that could influence a patient's kinesiophobia. If possible data on the disease duration, occupation, marital status, tobacco and alcohol consumption, level of education, personal background, and family histories, so morbidities as major depression and medications should ideally be included. As similar indicators like CRP, a fundamental disease activity measure, would have been suitable to include. Again, the presence of syndesmophytes or BASMI measurements indicating the status of spinal involvement could have been included.
Were all patients diagnosed with AS who visited the outpatient clinic during the specified time frame included? Provide more information on the study development process. Clarify if there were any inclusion or exclusion criteria for participants.
Regression analyses could have been applied in methods of statistics.

Results:
The Table 1 and 2 transmits very little information, hence it was assumed that the information could be conveyed without a table.

Discussion:

As mentioned in the study, there are several studies in the literature that investigate the association between AS and kinesiofobia It is also clear that the number of patients in some of these is large (as Oskay et al). Please include in this part the areas where your study differs and is stronger than other studies in the literature.

Experimental design

Abstract:
In the methods section of the abstract the explanations of the abbreviations as BASDAI, VAS, BASFI, TSK should be included because they used for the first time.
Introduction:
The introduction section is too long. In the first paragraph of the introduction please focus on the AS, mobility, dysfunction and kinesiofobia instead of the long general information. The sentence structures in the introduction part should be evaluated, and numerous repeated conjunctions should be avoided. The study's goal should be thoroughly and clearly explained in the final section of the Introduction. Concentrate on important research and data that bolster the objectives of the study directly.
Methods:
The potential impact of the work is reduced by concerns for some methodological rigor. There are very few demographic data that could influence a patient's kinesiophobia. If possible data on the disease duration, occupation, marital status, tobacco and alcohol consumption, level of education, personal background, and family histories, so morbidities as major depression and medications should ideally be included. As similar indicators like CRP, a fundamental disease activity measure, would have been suitable to include. Again, the presence of syndesmophytes or BASMI measurements indicating the status of spinal involvement could have been included.
Were all patients diagnosed with AS who visited the outpatient clinic during the specified time frame included? Provide more information on the study development process. Clarify if there were any inclusion or exclusion criteria for participants.
Regression analyses could have been applied in methods of statistics.

Results:
The Table 1 and 2 transmits very little information, hence it was assumed that the information could be conveyed without a table.

Discussion:

As mentioned in the study, there are several studies in the literature that investigate the association between AS and kinesiofobia It is also clear that the number of patients in some of these is large (as Oskay et al). Please include in this part the areas where your study differs and is stronger than other studies in the literature.

Validity of the findings

Abstract:
In the methods section of the abstract the explanations of the abbreviations as BASDAI, VAS, BASFI, TSK should be included because they used for the first time.
Introduction:
The introduction section is too long. In the first paragraph of the introduction please focus on the AS, mobility, dysfunction and kinesiofobia instead of the long general information. The sentence structures in the introduction part should be evaluated, and numerous repeated conjunctions should be avoided. The study's goal should be thoroughly and clearly explained in the final section of the Introduction. Concentrate on important research and data that bolster the objectives of the study directly.
Methods:
The potential impact of the work is reduced by concerns for some methodological rigor. There are very few demographic data that could influence a patient's kinesiophobia. If possible data on the disease duration, occupation, marital status, tobacco and alcohol consumption, level of education, personal background, and family histories, so morbidities as major depression and medications should ideally be included. As similar indicators like CRP, a fundamental disease activity measure, would have been suitable to include. Again, the presence of syndesmophytes or BASMI measurements indicating the status of spinal involvement could have been included.
Were all patients diagnosed with AS who visited the outpatient clinic during the specified time frame included? Provide more information on the study development process. Clarify if there were any inclusion or exclusion criteria for participants.
Regression analyses could have been applied in methods of statistics.

Results:
The Table 1 and 2 transmits very little information, hence it was assumed that the information could be conveyed without a table.

Discussion:

As mentioned in the study, there are several studies in the literature that investigate the association between AS and kinesiofobia It is also clear that the number of patients in some of these is large (as Oskay et al). Please include in this part the areas where your study differs and is stronger than other studies in the literature.

Additional comments

Abstract:
In the methods section of the abstract the explanations of the abbreviations as BASDAI, VAS, BASFI, TSK should be included because they used for the first time.
Introduction:
The introduction section is too long. In the first paragraph of the introduction please focus on the AS, mobility, dysfunction and kinesiofobia instead of the long general information. The sentence structures in the introduction part should be evaluated, and numerous repeated conjunctions should be avoided. The study's goal should be thoroughly and clearly explained in the final section of the Introduction. Concentrate on important research and data that bolster the objectives of the study directly.
Methods:
The potential impact of the work is reduced by concerns for some methodological rigor. There are very few demographic data that could influence a patient's kinesiophobia. If possible data on the disease duration, occupation, marital status, tobacco and alcohol consumption, level of education, personal background, and family histories, so morbidities as major depression and medications should ideally be included. As similar indicators like CRP, a fundamental disease activity measure, would have been suitable to include. Again, the presence of syndesmophytes or BASMI measurements indicating the status of spinal involvement could have been included.
Were all patients diagnosed with AS who visited the outpatient clinic during the specified time frame included? Provide more information on the study development process. Clarify if there were any inclusion or exclusion criteria for participants.
Regression analyses could have been applied in methods of statistics.

Results:
The Table 1 and 2 transmits very little information, hence it was assumed that the information could be conveyed without a table.

Discussion:

As mentioned in the study, there are several studies in the literature that investigate the association between AS and kinesiofobia It is also clear that the number of patients in some of these is large (as Oskay et al). Please include in this part the areas where your study differs and is stronger than other studies in the literature.

Annotated reviews are not available for download in order to protect the identity of reviewers who chose to remain anonymous.
Cite this review as

---

## Round 0.3 · Major Revisions

We have one reviewer who has commented "Very basic data were used in the study's design. The data do not include any key factors that could explain the kinesiophobia-AS association. The author claimed not to have these data when asked in revision".

Can you please either explain the kinesiophobia-AS association with the data you have or write justification on how the existing paper makes a contribution to the scientific literature in this area.

Thanks, A/Prof M Climstein

Reviewer 4 ·

Basic reporting

The grammar and language features used in the study were considered sufficient. The introduction part of the study is arranged in sufficient detail. The hypotheses are included in an adequate and explanatory manner.

Experimental design

The method part of the study was designed very simply. The lack of some basic data that may affect the results was reported by the author as a limitation (as lack of BASMI).

Validity of the findings

The study was conducted with a small number of patients and the presence of similar data in the past literature stands out. In this respect, the originality of the article can be questioned.

There is no comment on the cause-effect relationship in the discussion. For example, a connection between disease activity and kinesiophobia is included, but does kinesiophobia cause active disease or does kinesiophobia develop in AS patients with active disease? This issue remains open.

Cite this review as

---

## Round 0.4 · accepted · Accept

Authors, thank you for making the minor amendments to your manuscript. I am pleased to inform you that I am recommending your amended manuscript for publication. Thanks, A/Prof Mike Climstein (Section Editor, PeerJ)